# Intermittent antegrade warm-blood versus cold-blood cardioplegia in children undergoing open heart surgery: a protocol for a randomised controlled study (Thermic-3)

Rachael Heys [1,2] Serban Stoica,[3] Gianni Angelini,[2,4] Richard Beringer,[3] Rebecca Evans,[1,2] Mohamed Ghorbel,[4] William Lansdowne,[3] Andrew Parry,[3] Guido Pieles,[3] Barnaby Reeves [1,2] Chris Rogers,[1,2] Rohit Saxena,[5] Karen Sheehan,[2,3] Stella Smith,[3] Terrie Walker-Smith,[1] Robert MR Tulloh,[2,3,4] Massimo Caputo[3]

RH and SS contributed equally.

For numbered affiliations see end of article.

**Correspondence to**
Rachael Heys;
rh13369@bristol.ac.uk

## ABSTRACT

**Introduction** Surgical repair of congenital heart defects often requires the use of cardiopulmonary bypass (CPB) and cardioplegic arrest. Cardioplegia is used during cardiac surgery requiring CPB to keep the heart still and to reduce myocardial damage as a result of ischaemia–reperfusion injury. Cold cardioplegia is the prevalent method of myocardial protection in paediatric patients; however, warm cardioplegia is used as part of usual care throughout the UK in adults. We aim to provide evidence to support the use of warm versus cold blood cardioplegia on clinical and biochemical outcomes during and after paediatric congenital heart surgery.

**Methods and analysis** We are conducting a single-centre randomised controlled trial in paediatric patients undergoing operations requiring CPB and cardioplegic arrest at the Bristol Royal Hospital for Children. We will randomise participants in a 1:1 ratio to receive either 'cold-blood cardioplegia' or 'warm-blood cardioplegia'. The primary outcome will be the difference between groups with respect to Troponin T levels over the first 48 postoperative hours. Secondary outcomes will include measures of cardiac function; renal function; cerebral function; arrythmias during and postoperative hours; postoperative blood loss in the first 12 hours; vasoactive-inotrope score in the first 48 hours; intubation time; chest and wound infections; time from return from theatre until fit for discharge; length of postoperative hospital stay; all-cause mortality to 3 months postoperative; myocardial injury at the molecular and cellular level.

**Ethics and dissemination** This trial has been approved by the London – Central Research Ethics Committee. Findings will be disseminated to the academic community through peer-reviewed publications and presentation at national and international meetings. Patients will be informed of the results through patient organisations and newsletters to participants.

**Trial registration number** ISRCTN13467772; Pre-results.

## Strengths and limitations of this study

► This is the first randomised controlled trial to compare warm-blood cardioplegia with cold blood cardioplegia in the paediatric cardiac population. Previous research has compared cold crystalloid cardioplegia with warm-blood cardioplegia, so we hope to provide evidence to support the use of one type of cardioplegia.

► We are incorporating the Trials Engagement in Children and Adolescents Study Within A Trial. We will use trial data to answer multiple objectives, informing evidence-based decisions in trials.

► We have a broad eligibility criterion, so the trial results should be generalisable to a large population, including both simple and complex cardiac surgery procedures.

► Stratification will be used to ensure that surgical complexity is balanced across the two treatment groups.

► The primary outcome and majority of secondary outcomes are objective. Staff providing postoperative care will be blinded to the treatment allocation to minimise performance and detection bias.

## INTRODUCTION
### Background
Surgical repair of congenital heart defects often requires the use of cardiopulmonary bypass (CPB). CPB is a technique that temporarily takes over the function of the heart and lungs (pumping blood and oxygen through the body) and occasionally involves stopping the heart during surgery. In cases where the heart must be stopped, use of CPB is associated with complications such as myocardial damage as a result of ischaemia–reperfusion injury

(IRI; myocardial damage occurs when the blood supply returns to the heart after a period of restricted blood flow/ischaemia). Cardioplegia is used to stop the heart and for myocardial protection during cardiac surgery requiring CPB. Recovery and outcome statistics show a continued need to improve myocardial protection: myocardial damage can prolong hospital stay and result in delayed myocardial fibrosis[1]; and myocardial damage with low cardiac output is the most common cause of morbidity after surgical repair.[1–4]

Currently, there is wide variation in the use of cardioplegia in congenital cardiac surgery. Hypothermic cardioplegia solution is the prevalent method of myocardial protection in paediatric patients, however, there is wide variation in practice.[5 6] Cold cardioplegia is used to lower myocardial oxygen demands and the risk of IRI. However, intermittent warm-blood cardioplegia (IWBC) is considered to be an effective method of myocardial protection in adults undergoing surgical repair and is used in usual practise in the UK. The main benefits can be summarised as: (1) improved oxygen supply and reduced myocardial oedema; (2) more practical, because a simple electrically driven pump can be used instead of more complex cooling devices; and (3) cost-effective, because the expenses for cooling devices are not required.[7]

There have been a series of clinical and experimental studies confirming the effectiveness of IWBC compared with cold-blood cardioplegia (CBC) in adults.[8 9]

Findings from the adult cardiac surgery field cannot be applied to paediatric cardiac surgery patients. The first studies looking at the effectiveness of IWBC in children undergoing cardiac surgery have demonstrated that IWBC is a safe and effective strategy of myocardial protection. However, the studies have been limited to a retrospective comparison of 1400 patients who underwent cardiac surgery using IWBC with 950 patients treated with intermittent cold-bloodcardioplegia (ICBC)[10]; and a small single centre randomisedcontrolled trial (RCT) which compared cold crystalloid cardioplegia with IWBC.[11]

We would like to take this a step further and provide evidence to support the use of one type of cardioplegia[12] by investigating the effects of warm versus CBC on clinical and biochemical outcomes after paediatric congenital heart surgery.

### Embedded TRECA Study Within A Trial (SWAT)

The Thermic-3 trial is embedding the Trials Engagement in Children and Adolescents (TRECA) SWAT.[13] This SWAT is investigating a novel method for recruitment to see whether providing children and their families with information about a trial through multimedia information (MMI) resources impacts on recruitment and retention rates and the quality of decision-making about trial participation. The MMI resources are websites with text, images, animations and videos about the Thermic-3 trial. They have been developed by TRECA in conjunction with the Thermic-3 trial team.

### Aims and objectives

The main aim of the Thermic-3 trial is to compare IWBC with ICBC in paediatric patients undergoing open heart surgery. The objectives are to estimate the differences between groups with respect to the following:
1. Myocardial damage using serum Troponin T (cTnT) levels (primary outcome).
2. A range of clinical postsurgical outcomes: cardiac function; cardiac index; renal function; arrythmias intraoperatively and postoperatively; postoperative blood loss in the first 12 hours; vasoactive-inotrope score (VIS) in the first 48 hours; intubation time; chest and wound infections; time from return from theatre until fit for discharge from paediatric intensive care unit; length of postoperative hospital stay; all-cause mortality to 3 months postsurgery; myocardial injury at the molecular and cellular level (secondary outcomes).

## METHODS AND ANALYSIS
### Trial design and population

The Thermic-3 trial is a single-centre RCT comparing warm-blood cardioplegia (WBC) with CBC. We will recruit patients being referred for cardiac surgery requiring CPB and cardioplegia arrest at the Bristol Royal Hospital for Children (BRHC). Thermic-3 recruitment began in May 2018. The first patient was randomised on 2 June 2018. We anticipate that recruitment will be completed within 24 months.

### Eligibility criteria

Patients will be eligible if they:
1. Are undergoing a congenital heart operation requiring CPB and cardioplegic arrest at the BRHC (age range 0–18 years old).

Patients will be ineligible if they:
1. Weigh <3 kg.
2. Require an emergency operation (patient with haemodynamic instability who require immediate surgical intervention defined as operation within 24 hours of admission).
3. Require secundum atrial septal defect repair as an isolated procedure.
4. Are judged preoperatively by the surgeon to require deep hypothermic circulatory arrest (eg, aortic arch repair, repair of total anomalous pulmonary venous drainage, Norwood procedure).
5. Are judged pre-operatively by the surgeon to require deep hypothermic CPB.
6. Have a procedure that is judged by the surgeon to be preoperatively too complex. This could be procedure related or patient related (eg, complex tailor-made surgery; necrotising enterocolitis; preoperative brain haemorrhage; or generalised bleeding state/ongoing major bleeding).
7. Are of consenting/assenting age, however lacking capacity to consent/assent.

8. Are under the care of social services and/or the parent/guardian is unavailable for consent.

## Patient approach and consent

Potential trial participants will be identified from theatre and clinic lists. Prior to approach about the Thermic-3 trial, potential patients will be randomised (1:1:1) to one of the three TRECA SWAT groups: patient information leaflet (PIL), MMI or both PIL and MMI.[13] Parents/guardians will be given either a PIL, MMI or both PIL and MMI. Patients aged over 7 years old will also be provided with a PIL, MMI or both PIL and MMI suitable to their age group (7–10 years, 11–15 years or 16 years and over). If the study information is posted, parents/guardians or patients will also receive an invitation letter. All invitation letters, PILs and MMI have been approved by the Research Ethics Committee (REC). The MMI will be provided in the same way as the PILs. For example, the research nurse will be able to show the family the MMI on a tablet at the hospital. All parents/guardians and patients allocated to receive MMI will receive the link to the MMI on a laminated card so that they can review the MMI at home, either on a computer, tablet or smart phone.

Most patients will have at least 24 hours to consider whether to participate. In a few cases, this time interval may be less, for example, patients admitted for an operation without prior notification. Despite short notice, it is important to include these patients for the applicability of the trial findings. Written informed consent/assent for Thermic-3 will be obtained from the patient and/or their parent/guardian (where appropriate) prior to inclusion. Online supplemental files 1–3 show the consent/assent forms for the Thermic-3 trial. Figure 1 shows the expected patient pathway for the Thermic-3 trial.

## Trial interventions

Eligible patients who give consent will be randomised to receive one of the following:

1. WBC: blood cardioplegia at the same temperature as the body (≥34°C).
2. CBC: blood cardioplegia at 4°C–6°C.

For both groups, route of cardioplegia infusion will usually be into the aortic root, or selectively into the right or left coronary artery. The cardioplegia delivery methods will differ between the two groups. First, the temperature of the cardioplegia will be delivered as per the patient's randomisation allocation (warm or cold). Second, the delivery for the cardioplegia will differ between groups. For example, the concentration of the cardioplegia and frequency of infusion and reinjection rates differ slightly between groups. Details of the differences are described below. For the purpose of the trial, the two different techniques are abbreviated as 'warm' and 'cold' cardioplegia. The cardioplegia concentrations are provided in online supplemental file 4.

### Warm cardioplegia

The cardioplegic solution will be prepared by mixing a patient's own blood with sterile concentrate for cardioplegia infusion to achieve a final composition of 20 mM $K^+$ and 5 mM $Mg^{2+}$.[10] Blood will be withdrawn directly from the oxygenator using a pump and then re-infused at ≥34°C.

Initial infusion and reinjection rates will be calculated according to the patient's body surface area. The flow will be equal to 1–1.5 times the physiologic coronary flow

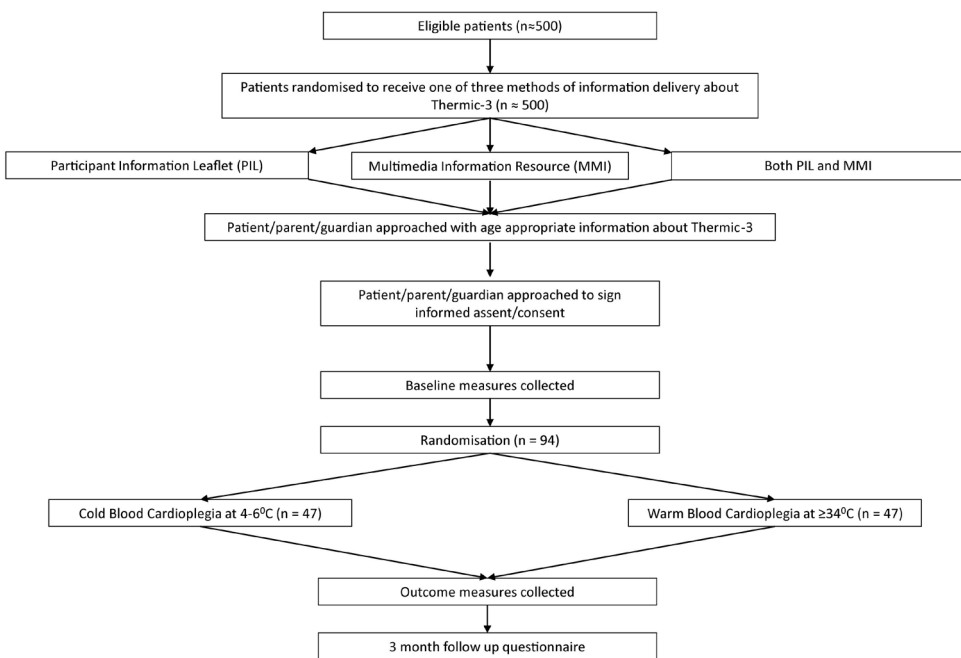

**Figure 1** Trial schema showing the recruitment pathway with the number of patients to be recruited, anticipated eligibility and recruitment rates.

(estimated as 5% of the cardiac output). The arresting dose of cardioplegia will be given for 1 min following electromechanical arrest of the heart. Reinjections at two-thirds of the initial speed of injection will be performed for 1 min every 15 min during aortic cross-clamping.

## Cold cardioplegia

Cold blood (4°C–6°C) high strength cardioplegic solution will be used for myocardial preservation. This will be mixed with the patients' blood withdrawn from the CPB circuit to give a ratio of 4 parts blood to 1-part cardioplegia solution. The mixture will be administered via a cardioplegia module that cools the solution to a temperature of approximately 4°C.

The induction dose will be $110 \, mL/min/m^2$, antegradely for 4 min, with a maintenance dose of $110 \, mL/min/m^2$ for 2 min at 20/30 min intervals.

The CPB temperature will be at the discretion of the operating surgeon. This will allow a degree of adaptation depending on the operation type. Temperature data will be collected throughout the operation to have a good understanding of the temperature variation on CPB.

## Randomisation

Eligibility will be confirmed, and consent given before randomisation takes place. The sequence of random allocations will be prepared in advance by a statistician using a computer by block randomisation method, with varying block sizes. Randomisation will take place via a secure password-protected database. Randomisation allocation will be concealed from all clinical and research personnel until a patient has been recruited and registered onto the trial database. Participants will then be randomly assigned in a 1:1 ratio and will be stratified by surgical complexity (Risk Adjustment for Congenital Heart Surgery (RACHS) score <3; RACHS score ≥3).

Randomisation within Thermic-3 will take place as close to the start of surgery as possible. If a participant's surgery is unexpectedly rescheduled, the trial number and randomisation allocation will be retained.

## Trial outcomes

The primary outcome will be cTnT levels over the first 48 postoperative hours (measured using Roche Elecsys Troponin T-hs (TnT-hs) assay, the routine analysis method used by the recruiting hospital). cTnT will be measured presurgery and at 2 hours, 6 hours, 24 hours and 48 hours post cross-clamp (XC) removal.

The secondary outcomes include the following:
1. Cardiac function assessed by indirect measures of cardiac output (central venous saturations ($ScvO_2$), arterial saturations, base deficit and blood lactate) will also be recorded. $ScvO_2$ will be measured via venous blood aspirated from the central venous catheter, up to 48 hours after XC removal. Arterial saturations, base deficit and lactate will be measured in arterial blood via sampling from the peripheral arterial cannula.

2. Routine blood gas and blood samples (pH, $PO_2$, $PCO_2$, base excess, lactate, C reactive protein, haemoglobin, haematocrit, white cell count and alanine aminotransferase).
3. Measures of renal function (urinary albumin, urinary creatinine, retinal binding protein, N-acetyl-β-glucosaminidase and neutrophil gelatinase-associated lipocalin).
4. Arrhythmias including atrial fibrillation/flutter, ventricular tachycardia, ventricular fibrillation, nodal, junctional ectopic tachycardia (JET) or atrioventricular (AV) block.
5. Postoperative blood loss in the first 12 hours.
6. New onset of arrhythmia postoperatively (either supraventricular tachycardia/atrial fibrillation or ventricular fibrillation/ventricular tachycardia, JET or heart block).
7. VIS over the first 48 hours after admission to paediatric intensive care unit (PICU).[14]
8. Postoperative intubation time in hours (including instances of reintubation).
9. Time from return from theatre until fit for transfer to the ward/high dependency unit (HDU) from the PICU. Fit for discharge to the ward/HDU will reflect current ward/HDU admission criteria.
10. Chest and wound infections.
11. All-cause mortality to 3 months postsurgery.
12. Length of postoperative hospital stay in days (from surgery to discharge from hospital).
13. Myocardial injury at the molecular and cellular level (eg, transcriptomics, proteomics and metabolomics)[15–17] measured in samples considered clinical waste. Where sufficient waste tissue, plasma and serum are available, metabonomics, proteonomics and transcriptomics, microRNA and isolated progenitor cells will be investigated. It will be assessed whether the isolated progenitor cells have similar regenerative capacity, in patients subjected to different surgical protocols.

## Data collection

The schedule of data collection is shown in table 1. Data will be collected from medical notes and hospital records, entered onto a bespoke database and stored on a secure server.

## Blinding

Staff providing postoperative care will be blinded to the participant's treatment allocation. Documentation containing the treatment allocation (eg, perfusion charts) will be placed in a sealed envelope within the medical notes. Any member of staff accessing the documents will be asked to record their name and the reason for accessing the documentation.

## Sample size calculation

The sample size calculation is based on cTnT levels which were measured in a previous RCT.[16] The data were log

**Table 1** Schedule of data collection

| Data items and samples collected | Preoperative | Postoperative | | | | | | | | | | | | | 3 Months postoperative |
|---|---|---|---|---|---|---|---|---|---|---|---|---|---|---|---|
| | | Presurgery | Start of CPB | 10min post CPB | XC removal | XC +1 hour | XC +2 hours | XC +4 hours | XC +6 hours | XC +12 hours | XC +24 hours | XC +48 hours | Subsequent days up to hospital discharge | Hospital discharge | |
| Screening and consent data | ✓ | | | | | | | | | | | | | | |
| Baseline data (patient demographics, medical history, medications) | | ✓ | | | | | | | | | | | | | |
| Randomisation allocation | ✓ | | | | | | | | | | | | | | |
| Routine blood gases | | ✓ | | | ✓ | ✓ | ✓ | | | ✓ | ✓ | ✓ | | | |
| Routine blood samples | | ✓ | | | | | ✓* | | | | ✓ | | ✓ | | |
| Blood for Troponin T | | ✓ | | | | | ✓ | | ✓ | ✓ | | ✓ | | | |
| Central venous saturations | | | | | | ✓† | ✓† | ✓† | ✓† | ✓† | ✓† | ✓† | | | |
| Arterial saturations, base deficit and lactate | | ✓ (induction) | | | | ✓ | ✓ | | ✓ | | ✓ | | | | |
| Urinary albumin and creatinine, RBP, NAG and N-GAL | | ✓ | | | | | ✓ | | | ✓ | ✓ | | | | |
| Operative details‡ | | ✓ | ✓ | ✓ | ✓ | | | | | | | | | | |
| Waste heart tissue (if available) | | | | | ✓ | | | | | | | | | | |
| Clinical outcomes | | | | | | | | | | | | | | ✓ | |
| Safety data postdischarge | | | | | | | | | | | | | | | ✓ |

*Routine bloods taken on admission to PICU.
†Several ScvO₂ readings will be taken from XC removal until XC+ 48 hours.
‡Operative details are recorded on XC removal and chest closure.
CPB, cardiopulmonary bypass; NAG, N-acetyl-β-glucosaminidase; N-GAL, neutrophil gelatinase-associated lipocalin; PICU, paediatric intensive care unit; RBP, retinal binding protein; XC, cross-clamp.

normal with correlations between pre and post measures of about 0.3 and between successive post measures of about 0.5. Given these correlations and four repeated postoperative measures, a sample size of 94 patients (47 per group) is sufficient to detect a standardised difference of 0.46 SDs (moderate effect size) in cTnT with 80% power and 5% significance (two-tailed), allowing for up to 15% missing data for cTnT.

## Statistical analysis

The trial will be analysed on an intention-to-treat basis, that is, outcomes will be analysed according to the treatment allocation, following a prespecified analysis plan. Any non-adherence to the allocated group will be documented. The primary outcome (cTnT levels over the first 48 postoperative hours) will be analysed using a longitudinal mixed-regression model, which allows for unbalanced data. All secondary outcomes will yield either binary, quantitative, time to event or longitudinal data, and will be analysed using linear regression (quantitative outcomes) or survival methods (time to event outcomes). Binary secondary outcomes will be described but not formally compared. Analyses will be adjusted for baseline values (where measured), and surgical complexity (stratification factor). Differences between the two groups will be reported as effect sizes with 95% CIs.

The transcriptomics data will be analysed using specialised software. Gene expression profiles will be determined, then the biological function of the identified genes will be established in order to generate hypotheses about their role. The signalling pathway to which these genes belong will be explored. The selection of important candidate genes will be validated.

It is expected that low numbers of adverse events are likely to be experienced by trial participants, based on our previous trials and due to the relatively small sample size and short follow-up period. For this reason the trial will not have an independent Data Monitoring and Safety Committee, but will be overseen by the Steering Group established to provide oversight of all studies supported by the National Institute for Health Research Bristol Cardiovascular Biomedical Research Centre Cardiovascular theme.

## Patient and public involvement (PPI)

The trial information leaflets for paediatric patients (7–10, 11–15 and 16–17 years) have been reviewed by the Generation R Young People's Advisory Group (YPAG) at the University Hospitals Bristol NHS trust. The parent and guardian information leaflets have been reviewed by a parent and guardian group, part of the Cardiovascular Biomedical Research Centre (CV-BRC) PPI advisory group. Changes to the information leaflets were made based on the feedback received, to ensure the content and format of the PILs were acceptable to the target audience.

Patients will be informed of the results through patient organisations and a summary report for participants. The results summary for participants will be designed with input from the CV-BRC and YPAG PPI advisory groups.

## RISK OF BIAS

The following features of the trial design will minimise the risk of bias. To minimise selection bias, stratification will be used to ensure that surgical complexity is balanced across the two groups and the treatment allocation will only be disclosed to the research and clinical team after patient eligibility and consent has been confirmed.

To minimise performance bias the warm and cold cardioplegia groups are defined in the trial protocol, as well as standard protocols for procedures undertaken during the trial (eg, perfusion and anaesthetic guidelines). Additionally, staff providing post-operative care will be blinded to the treatment allocation and this will also help minimise detection bias.

We expect minimal missing outcome data, as most data will be collected during the participant's hospital visit. Furthermore, missing data should not be differential across groups. The follow-up period for the trial is short (3 months), so we expect lost to follow-up to be small. We will follow-up participants (eg, contacting participants with overdue questionnaires) to maximise questionnaire completion rates.

## ETHICS AND DISSEMINATION

The trial received research ethics approval from the London - Central Research Ethics Committee (REC) in March 2018. The trial is managed by the Bristol Trials Centre, Clinical Trials and Evaluation Unit (CTEU) and sponsored by University Hospitals Bristol NHS Foundation Trust (www.uhbristol.nhs.uk/research-innovation/). Participants and their parents/guardians (where applicable) have the right to withdraw at any time and if they do withdraw, will be treated according to hospital standard procedures. Participants who choose to withdraw from the trial will be asked if we can continue to use any data already collected and whether they are willing to participate in the trial follow-up. We will present the trial findings at international meetings and peer-reviewed publications. We will inform the public through patient organisations and a newsletter to participants.

### Changes to protocol since REC approval

Since the trial received ethics approval, we have made three significant changes to the trial protocol. The first substantial amendment was undertaken to incorporate the TRECA SWAT into the Thermic-3 trial; to allow the use of MMI resources in the recruitment process for the Thermic-3 trial. The protocol was also updated to amend the cardioplegia manufacturer used in the WBC group, to reflect current practise in BRHC.

The second substantial amendment was undertaken to update that warm cardioplegia is used as part of standard practise in the BRHC. The protocol and trial documents were amended.

The third substantial amendment was undertaken to update the secondary outcomes for the study. The changes have been made to remove the secondary outcomes; transthoracic and transoesophageal echocardiography; and

cerebral function/injury as measured by Glial Fibrillar Acidic Protein (GFAP).

We decided to remove the collection of transthoracic and transoesophageal echocardiography data due to issues collecting the data. The data collection was not deliverable due to time constraints and difficultly collecting data in surgery.

The GFAP data collection was removed as a significant number of results were below the limit of detection and there was a high variability between replicates.

Version 7.0 (dated 7 August 2019) of the protocol is currently in use. The relevant regulatory approvals will be obtained for amendments to the protocol and trial information leaflets. Relevant parties (eg, research team members, research nurses) will be informed via email. If changes are made to trial processes, participants will be asked to reconsent if necessary.

**Author affiliations**
[1]Bristol Trials Centre, Clincal Trials and Evaulation Unit, Bristol Medical School, University of Bristol, Bristol, UK
[2]National Institute for Health Research Bristol Biomedical Research Centre, University Hospitals Bristol NHS Foundation Trust and University of Bristol, Bristol, UK
[3]Bristol Royal Hospital for Children, University Hospitals Bristol NHS Foundation Trust, Bristol, UK
[4]Bristol Heart Institue, University of Bristol, Bristol, UK
[5]Cardiac Intensive Care, Great Ormond Street Hospital For Children NHS Foundation Trust, London, UK

**Acknowledgements** The Thermic-3 trial is sponsored by University Hospitals Bristol NHS Foundation Trust. The sponsor is responsible for the oversight of the Thermic-3 trial and will ensure the trial is managed appropriately. This trial was designed and delivered in collaboration with the Bristol Trials Centre (CTEU), a UKCRC registered clinical trials unit, which is in receipt of National Institute for Health Research CTU support funding. The research team acknowledges the support of the National Institute for Health Research Clinical Research Network (NIHR CRN). The authors would like to thank all trial team members involved in the recruitment, coordination and data entry for this trial and members of the YPAG and CV-BRC PPI group and public who assisted with the review of trial documents. Special thanks are also given to perfusion, anaesthetics and nursing teams for their support in the conduct of the trial.

**Contributors** RH helped in preparing and drafting the trial protocol and writing the manuscript. SS is the chief investigator who helped in trial design, preparation of trial protocol, definition of trial interventions, definition of outcomes and writing the manuscript. RE helped in the preparation and drafting of trial protocol, sample size and statistical analysis plan and review of manuscript. AP is a participating surgeon in the trial and performed the review of the manuscript. BR helped in trial design, preparation and drafting of trial protocol and review of the manuscript. CR helped in trial design, preparation and drafting of trial protocol, sample size and statistical analysis plan and review of the manuscript. SSm and TWS performed review of trial protocol and writing the manuscript. MC helped in trial concept, trial design, preparation of trial protocol, definition of trial interventions, definition of outcomes, training of perfusionists and review of the manuscript. GA, RB, MG, WL, GP, RS, KS and RMRT performed trial design, review of trial protocol and review of the manuscript. All authors read and approved the final manuscript.

**Funding** The trial is supported by the National Institute for Health Research (NIHR) Bristol Biomedical Research Centre (Cardiovascular theme) and funded by the British Heart Foundation (CH/1992027/7163 and CH/17/1/32804). The views and opinions expressed herein are those of the authors and do not necessarily reflect those of the NIHR, NHS or the Department of Health. The NIHR and BHF will not be involved in the trial conduct, analysis or reporting of the trial.

**Competing interests** None declared.

**Patient and public involvement** Patients and/or the public were involved in the design, or conduct, or reporting or dissemination plans of this research. Refer to the Methods section for further details.

**Patient consent for publication** Not required.

**Provenance and peer review** Not commissioned; externally peer reviewed.

**ORCID iDs**
Rachael Heys http://orcid.org/0000-0002-3957-1287
Barnaby Reeves http://orcid.org/0000-0002-5101-9487

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
