## [Reviewer comments · BMJ Open]

ARTICLE DETAILS

TITLE (PROVISIONAL)	Intermittent Antegrade Warm Blood versus Cold Blood Cardioplegia in Children Undergoing Open Heart Surgery; a protocol for a Randomised Controlled Study (Thermic-3)
AUTHORS	Heys, Rachael; Stoica, Serban; Angelini, Gianni; Beringer, Richard; Evans, Rebecca; Ghorbel, Mohamed; Lansdowne, William; Parry, Andrew; Pieves, Guido; Reeves, Barnaby; Rogers, Chris; Saxena, Rohit; Sheehan, Karen; Smith, Stella; Walker-Smith, Terrie; Tulloh, Robert; Caputo, Massimo

VERSION 1 - REVIEW

REVIEWER	Stephen Femes Sunnybrook Health Sciences Centre University of Toronto Canada
REVIEW RETURNED	01-Feb-2020

GENERAL COMMENTS	The study is funded and approved, so many of these comments may be viewed as irrelevant. The manuscript is very clearly written. Has the study started? If so, please include the date that the study began. There are protocol changes described, which could be interpreted to mean that the study has started. The study is a single centre study - inclusion of more centres would increase the generalizability of the findings of the study. Randomization: The mode of randomization is not described - i.e. sealed envelopes, computer interaction, etc. Study Outcomes: The authors have for the most part listed the outcomes, rather than provide definitions of the outcomes. The authors can include an appendix for the definitions of the study outcomes. There is a primary outcome, then a very lengthy list of secondary outcomes - it is unclear how the authors will control the study wise type 1 error. Primary outcome: The authors have chosen troponin as the primary study outcome. Troponin is a surrogate measure of cardiac injury - a
---

	clinical outcome would be preferable. The study is powered for a moderate effect size, but I believe that is a moderate effect size in a longitudinal mixed model of troponin measurements - it is likely that the estimates of mean troponin values over time may all be in a somewhat normal range of postoperative values. I agree that differences in the proportions of patients with very high values are likely clinically relevant. Are these high sensitivity troponins? The authors should include the assay for the test. I also suggest that the authors explore the association of the troponin results and clinical outcome. Sample size - the authors could include more information in the main text (or an appendix) about the derivation of the sample size. Analysis: There are 2 levels of randomization - it is unlikely that the TRECA interventions affect outcomes, but one can never know. The authors should describe the planned analysis incorporating the primary randomization and also warm vs cold cardioplegia. Page 13, Section m, and Page 15, Lines 53-60: The description of this section is cursory - also not clear from what is written whether there are some specific candidate markers that will be evaluated or a very large panel. The included references are to prior trials conducted by GA. Feasibility: What is the proposed timeline for patient recruitment? Is there a plan to include other pediatric cardiac centres if recruitment is slower than anticipated? Or do the authors have other changes to the inclusion/exclusion criteria they might envision if recruitment does not proceed as quickly as they expect? References: The references do not seem up to date, but may be up to date wrt this research question. On the other hand, the identification of myocardial injury perioperatively is very topical.
--	---

REVIEWER	Mr Nigel Drury University of Birmingham, UK
REVIEW RETURNED	04-Feb-2020

GENERAL COMMENTS	This manuscript reports the study protocol for a phase II, single-centre, randomised controlled trial of intermittent antegrade warm blood versus cold blood cardioplegia in children. Patient recruitment is underway and it is being conducted by experienced clinical investigators through a UKCRC-registered CTU. Whilst the manuscript is fairly well written, it lacks important detail in a number of areas such that with this information alone, the study could not be repeated. These include:  - Eligibility criteria: The age range for eligibility should be defined (p9/13-14). - Primary outcome: 'Troponin levels over the first 48 post-operative hours' (p8/39, p12/14-15). How is this measured/calculated? Is it area under the time-concentration curve, peak value, total release, or at specific time-points? What troponin assay is being used (eg. high-sensitivity) and on what analysis platform (ie. manufacturer).
---

The four post-op time-points are included in the schedule (table 1) but should also be defined in the text. The point from which these are timed from should also be defined – aortic cross-clamp removal, end of skin closure, arrival in PICU? This level of detail should also be included in the abstract.

- Secondary outcome: 'Cardiac function; assessed by measuring cardiac output and cardiac index' (p8/41, p12/21-24). There is no indication of how these will be measured; the use of transthoracic and transoesophageal echo were removed from the protocol in the third substantial amendment (p17/41-46) so is this still relevant?

- Secondary outcome: 'Intubation time' (p13/5). When is this measured from – aortic cross-clamp removal, end of skin closure, arrival in PICU? If the patient is extubated on the table, how will this be dealt with?

- Secondary outcome: 'Time from return from theatre until fit for transfer' (p13/8-10). The method for determining fitness for transfer should be documented.

Other points to be addressed:

- The authors claim several times that this trial aims to 'eliminate uncertainty' surrounding the use of warm blood cardioplegia (p5/19-20, p6/20, p8/7); this is a remarkably bold claim for a phase II trial and should be changed to 'provide evidence to support the use of', or similar.

- Abstract/strengths & limitations refers to the study cohort as a 'high-risk paediatric cardiac population' (p6/16) but their exclusion criteria include weight <3kg, emergency surgery, need for DHCA, and cases deemed 'too complex' by the surgeon. The target population therefore do not represent a 'high-risk' group within those children undergoing cardiac surgery and this term should be removed.

- Date of commencing the trial: The date on which the first patient was randomised should be included, as per the BMJ Open instructions for authors to include important trial dates.

- Terminology: 'cardioplegic application' would be better replaced with 'cardioplegic arrest' (p5/12, p9/3), and 'cardiac and electrical activity' (p5/37-38, p8/43, p12/41-45) should be more accurately termed 'arrhythmias' if this is what is meant.

- Table 1: No events are listed in the 'After 20 XC time' column – if this is correct, it should be removed. The superscript 3 does not correspond with the correct definition in the legend.

- DMSC: Despite the assertion that 'low numbers of adverse events are likely to be experienced by the trial participants' (p16/3-7), I was surprised that independent oversight by a DMSC was not deemed necessary by the investigators, sponsor, or funder for an interventional trial in children, although this practice remains commonplace in the literature.

- Funding statement: The specific grant numbers should be included (p19/50-60).

	- References: Many of the references are old with 1-6 all from the 1990s. Some more up-to-date references should be considered for variations in practice (Harvey B et al. J Extra Corpor Technol 2012;44:186-93, Kotani Y et al. Ann Thorac Surg 2013;96:923-9, Drury NE et al. Perfusion 2019;34:125-9), outcomes of cardioplegic arrest in children (Gaies M et al. Ann Thorac Surg 2016;102:628-35, Ma M et al. Ann Thorac Surg 2007;83:438-45), troponin as a marker of paediatric myocardial injury (Mildh LH et al. Ann Thorac Surg 2006;82:1643-9, Su JA et al. Semin Thorac Cardiovasc Surg 2019;31:244-51), and the current clinical trials literature (Drury NE et al. Interact Cardiovasc Thorac Surg 2019;28:144-150). This study raises no apparent ethical issues.
--	---

REVIEWER	Kuhn, Elmar University of Cologne, Germany
REVIEW RETURNED	13-Feb-2020

GENERAL COMMENTS	The authors provide a study protocol investigating the effect of warm versus cold cardioplegia in children undergoing various kinds of heart surgery. Here are the main aspects that need to be addressed:  - The null-hypothesis is unclear. What is supposed to perform better: warm or cold cardioplegia? What is the expected difference in troponin elevation? It remains unclear if the peak value or the area under the curve is included in the primary endpoint. Consequently, the included sample-size calculation is based on sparse evidence and needs to be fortified by additional data. If there is no data available, retrospective data analysis should be performed before. - The primary outcome is troponin T level over the first 48 h postoperative hours. The authors should add a comment why high-sensitive troponin is used instead. - The authors describe some exclusion criteria including the judgement of the surgeon. This is too subjective. If aortic arch repair surgery is associated with deep hypothermic arrest, children undergoing these procedures should be excluded without the judgment of the surgeon. Additionally, what is the definition of "deep" hypothermic CPB? Likewise, why not strictly exclude children with endocarditis? - The exact concentrations of both cardioplegic solutions need to be included. - The authors need to explain why "low numbers of adverse events" are likely to be detected.
--

VERSION 1 – AUTHOR RESPONSE

Reviewer: 1

1. Has the study started? If so, please include the date that the study began. There are protocol changes described, which could be interpreted to mean that the study has started. The recruitment start date has been added to the 'Trial design and population' on page 7 of the manuscript.

2. The study is a single centre study - inclusion of more centres would increase the generalizability of

the findings of the study. We agree, but we are not able to change this. We did investigate including more centres to increase the generalizability of the study findings, but for a variety of reasons we were not able to recruit additional centres in the time scale available. The main obstacles were logistical and not related to the importance of the research question.

3. Randomization: The mode of randomization is not described - i.e. sealed envelopes, computer interaction, etc. We apologise for omitting this detail, it is computer-based and this has been added to the 'Randomisation' Section on page 10 of the manuscript.

4. Study Outcomes: The authors have for the most part listed the outcomes, rather than provide definitions of the outcomes. The authors can include an appendix for the definitions of the study outcomes. The section 'Trial Outcomes' on page 10 and 11 of the manuscript has been updated with additional details of the definitions of the outcomes.

5. There is a primary outcome, then a very lengthy list of secondary outcomes - it is unclear how the authors will control the study wise type 1 error. There are differing opinions on the merits of controlling for or non-controlling for the type-1 error rate. This is reflected in the current CONSORT reporting guidelines for RCTs [BMJ 2010;340:c332] which do not make any recommendation in this regard. We do not plan to control for the type-1 error rate in this study. However, the number of statistical comparisons will be kept to a minimum and will be pre-specified in the statistical analysis plan and will not be applied for outcomes with low event rates and only pre-specified subgroup analyses will be performed. Consideration will be taken in interpretation of results to reflect the number of statistical tests performed and the consistency, magnitude, and direction of treatment estimates for different outcomes.

6. Primary outcome: The authors have chosen troponin as the primary study outcome. Troponin is a surrogate measure of cardiac injury - a clinical outcome would be preferable. The study is powered for a moderate effect size, but I believe that is a moderate effect size in a longitudinal mixed model of troponin measurements - it is likely that the estimates of mean troponin values over time may all be in a somewhat normal range of postoperative values. I agree that differences in the proportions of patients with very high values are likely clinically relevant. Are these high sensitivity troponins? The authors should include the assay for the test. I also suggest that the authors explore the association of the troponin results and clinical outcome.

We agree that a clinical outcome would be desirable, but such an outcome would be very difficult to

define and study, particularly in a small exploratory trial. A multicentre UK trial of 2 types of cardioplegia is starting imminently and this too has troponin as the primary outcome. Large trials with clinical endpoints are preferred, but in paediatric cardiac surgery they are difficult to achieve without international collaborations. Roche Elecsys® Troponin T-hs (TnT-hs) assay is being used for the study. The manuscript has been updated to include the assay for test, under section 'Trial Outcomes' on page 10 of the manuscript.

7. Sample size - the authors could include more information in the main text (or an appendix) about the derivation of the sample size. We have clarified the description of the sample size calculation as requested. We can confirm that the information provided is sufficient to allow the calculation to be reproduced.

8. Analysis: There are 2 levels of randomization - it is unlikely that the TRECA interventions affect outcomes, but one can never know. The authors should describe the planned analysis incorporating the primary randomization and also warm vs cold cardioplegia.

Potential participants will be randomised (1:1:1 ratio) to one of the three TRECA interventions prior to approach about the Thermic-3 trial. As this sub study will be prior to consent and randomisation for Thermic-3 (the main trial) participants should be balanced between Thermic-3 treatment groups and therefore we do not plan to adjust the main trial analysis for TRECA allocation. Additionally, clinical outcome assessors will be blinded to the TRECA allocation and the trial does not include patient-reported outcomes.

9. Page 13, Section m, and Page 15, Lines 53-60: The description of this section is cursory - also not clear from what is written whether there are some specific candidate markers that will be evaluated or a very large panel. The included references are to prior trials conducted by GA. Additional information has been added to the section 'Trial Outcomes' on page 11. Where sufficient waste tissue, plasma and serum are available, metabonomics, proteonomics and transcriptomics, microRNA, and isolated progenitor cells will be investigated. The aim will be to assess whether the isolated progenitor cells have similar regenerative capacity, in patients subjected to different surgical protocols.

10. Feasibility: What is the proposed timeline for patient recruitment? Is there a plan to include other pediatric cardiac centres if recruitment is slower than anticipated? Or do the authors have other changes to the inclusion/exclusion criteria they might envision if recruitment does not proceed

as quickly as they expect? Recruitment to the study began in May 2018 and we anticipate that recruitment will take 24 months. Section 'Trial design and population' on page 7 has been updated with the recruitment start date and proposed timeline. The study team considered the addition of another centre, but due to logistical issues it was decided that it would not be feasible to open an additional site in the remaining recruitment period.

11. References: The references do not seem up to date but may be up to date with this research question. On the other hand, the identification of myocardial injury perioperatively is very topical. The references have been reviewed and updated.

Reviewer: 2

1. Eligibility criteria: The age range for eligibility should be defined (p9/13-14). The study protocol does not specify the age range, patients undergoing a congenital heart operation requiring CPB and cardioplegic arrest at the Bristol Royal Hospital for Children will be eligible to participate, so the range is 0-18 years. This information has been added to the revised manuscript

2. Primary outcome: 'Troponin levels over the first 48 post-operative hours' (p8/39, p12/14-15). How is this measured/calculated? Is it area under the time-concentration curve, peak value, total release, or at specific time-points? What troponin assay is being used (eg. high sensitivity) and on what analysis platform (ie. manufacturer). The four post-op time-points are included in the schedule (table 1) but should also be defined in the text. The point from which these are timed from should also be defined – aortic cross-clamp removal, end of skin closure, arrival in PICU? This level of detail should also be included in the abstract. Roche Elecsys® Troponin T-hs (TnT-hs) assay is being used for

the study. The manuscript has been updated to include the assay for test, under section 'Trial Outcomes' on page 10. Space constraints do not allow the time-points for the primary outcome to be included in the abstract, but we have included this in the text in section 'Trial Outcomes' on page 10 of the manuscript. The cTnT concentrations measured at each time point will be compared between groups using longitudinal mixed regression as described in the statistical analysis section. Summary measures such as area under the curve/peak value will not be used.

3. Secondary outcome: 'Cardiac function; assessed by measuring cardiac output and cardiac index' (p8/41, p12/21-24). There is no indication of how these will be measured; the use of transthoracic and transoesophageal echo were removed from the protocol in the third substantial amendment

(p17/41-46) so is this still relevant? The section 'Trial Outcomes' on page 10 has been updated with further information on the measure of cardiac index being used.

4. Secondary outcome: 'Intubation time' (p13/5). When is this measured from – aortic cross-clamp removal, end of skin closure, arrival in PICU? If the patient is extubated on the table, how will this be dealt with? Post-operative intubation time will be measured (from knife down time until extubation, including instances of re-intubation). Patients extubated in theatre will be included (but this may be a short duration). Point 'H' of the section 'Trial Outcomes' has been updated.

5. Secondary outcome: 'Time from return from theatre until fit for transfer' (p13/8-10). The method for determining fitness for transfer should be documented. Fit for discharge to the ward/HDU will reflect current ward/HDU admission criteria. Point 'I' of the 'Trial Outcomes' has been updated.

6. The authors claim several times that this trial aims to 'eliminate uncertainty' surrounding the use of warm blood cardioplegia (p5/19-20, p6/20, p8/7); this is a remarkably bold claim for a phase II trial and should be changed to 'provide evidence to support the use of', or similar. The wording has been updated on p6/20 and p8/7. We agree, the text has been revised in line with the reviewer's suggestion.

7. Abstract/strengths & limitations refers to the study cohort as a 'high-risk paediatric cardiac population' (p6/16) but their exclusion criteria include weight <3kg, emergency surgery, need for DHCA, and cases deemed 'too complex' by the surgeon. The target population therefore do not represent a 'high-risk' group within those children undergoing cardiac surgery and this term should be removed. The manuscript has been updated to remove the term 'high-risk paediatric cardiac population' from the strengths and limitations (page 4) as suggested.

8. Date of commencing the trial: The date on which the first patient was randomised should be included, as per the BMJ Open instructions for authors to include important trial dates. The recruitment start date and date of first patient randomised has been updated in section 'Trial design and population' on page 7 of the manuscript.

9. Terminology: 'cardioplegic application' would be better replaced with 'cardioplegic arrest' (p5/12, p9/3), and 'cardiac and electrical activity' (p5/37-38, p8/43, p12/41-45) should be more accurately termed 'arrhythmias' if this is what is meant. The terminology has been updated as suggested; cardioplegic application has been replaced with cardioplegic arrest (p5/12 and p9/3) and 'cardiac and electrical activity' has been updated to 'arrhythmias' (p5/37-38, p8/43, p12/41-45).

10. Table 1: No events are listed in the 'After 20 XC time' column – if this is correct, it should be removed. The superscript 3 does not correspond with the correct definition in the legend. We thank the reviewer for highlighting this error; Table 1 the column 'After 20 min XC time' has been deleted – this does not show on the track changes. The superscripts for Table 1 have been checked and updated.

11. DMSC: Despite the assertion that 'low numbers of adverse events are likely to be experienced by the trial participants' (p16/3-7), I was surprised that independent oversight by a DMSC was not deemed necessary by the investigators, sponsor, or funder for an interventional trial in children, although this practice remains commonplace in the literature. The size of the trial, and short follow up period suggests that there will be a low number of expected events. Additionally, data from previous studies suggested a low adverse event rate. For this reason, it was decided that the trial will not have a DMSC. However, the study will be overseen by the Steering Group established to provide oversight of all studies supported by the National Institute for Health Research Bristol Cardiovascular Biomedical Research Centre Cardiovascular theme. The text has been updated in section 'Statistical Analysis' on page 14 to provide clarification.

12. Funding statement: The specific grant numbers should be included (p19/50-60). We thank the reviewers for highlighting this omission. The grant numbers have been included in the funding statement (page 20 of manuscript).

13. References: Many of the references are old with 1-6 all from the 1990s. Some more up-to-date references should be considered for variations in practice (Harvey B et al. *J Extra Corpor Technol* 2012;44:186-93, Kotani Y et al. *Ann Thorac Surg* 2013;96:923-9, Drury NE et al. *Perfusion* 2019;34:125-9), outcomes of cardioplegic arrest in children (Gaies M et al. *Ann Thorac Surg* 2016;102:628-35, Ma M et al. *Ann Thorac Surg* 2007;83:438-45), troponin as a marker of paediatric myocardial injury (Mildh LH et al. *Ann Thorac Surg* 2006;82:1643-9, Su JA et al. *Semin Thorac Cardiovasc Surg* 2019;31:244-51), and the current clinical trials literature (Drury NE et al. *Interact Cardiovasc Thorac Surg* 2019;28:144-150). Thank you for suggesting more up-to-date references. The background section has been reviewed and updated.

Reviewer: 3

1. The null-hypothesis is unclear. What is supposed to perform better: warm or cold cardioplegia? What is the expected difference in troponin elevation? It remains unclear if the peak value or the

area under the curve is included in the primary endpoint. Consequently, the included sample-size calculation is based on sparse evidence and needs to be fortified by additional data. If there is no data available, retrospective data analysis should be performed before. We thank the reviewer for the feedback. Cold cardioplegia is most prevalent in paediatric cardiac surgery but there is evidence in adult cardiac surgery that warm cardioplegia is beneficial and we are aiming to determine if there is evidence to support its use in paediatric cardiac surgery. The comments about the statistical analysis and sample size have been addressed above, as these points were also raised by reviewer 2.

2. The primary outcome is troponin T level over the first 48 h postoperative hours. The authors should add a comment why high-sensitive troponin is used instead. The Roche Elecsys® Troponin T (hs) assay is the standard troponin T assay used in the Trust where the study is being conducted. A comment has been added to the manuscript (page 10/15-16).

3. The authors describe some exclusion criteria including the judgement of the surgeon. This is too subjective. If aortic arch repair surgery is associated with deep hypothermic arrest, children undergoing these procedures should be excluded without the judgment of the surgeon. Additionally, what is the definition of "deep" hypothermic CPB? Likewise, why not strictly exclude children with endocarditis? The definitions for exclusion criteria try to find the middle road between pragmatism, ethics, and trial constraints. First, we did not think it makes sense to use warm cardioplegia when cold bypass is the main perfusion strategy; in this context deep hypothermia refers to CPB around 18-24°C. There are other cases where the temperature will be dropped briefly at the discretion of the surgeon, for example to be able to reduce flow if needed and improve the view. If this is done briefly and in unplanned fashion, we did not feel that it is an instance that warrants exclusion. Apart from these well-defined situations there is a small element of surgical discretion. One surgeon will feel comfortable to try warm cardioplegia in an arterial switch procedure, for example, another surgeon may want to stick to the well tested technique of cold cardioplegia and hypothermic bypass. To ensure good enrolment of consecutive consenting patients we wanted to make it as easy as possible for all surgeons to adapt their technique to the trial. By and large this has been very successful, and the discretionary exclusions have been minimal. In the same context we did not feel that endocarditis per se is a condition sufficiently precise to warrant exclusion. Hypothetically, some patients in the easy end of the spectrum head for a simple valve repair or replacement. There are however complex cases which require prolonged and difficult reconstructions. These are perhaps

circumstances when the operating surgeon does not need the added burden of a technique that is not routine, so this is an example where the surgeon is entitled to say no.

4. The exact concentrations of both cardioplegic solutions need to be included. The cardioplegia concentrations have been added in the supplementary material for the manuscript.

5. The authors need to explain why "low numbers of adverse events" are likely to be detected. The size of the trial, and short follow up period suggests that there will be a low number of expected events. Additionally, data from previous studies suggested a low adverse event rate. For this reason, it was decided that the trial will not warrant an independent DMSC. The text has been updated in section 'Statistical Analysis' on page 14 to provide clarification. However, the study will be overseen by the Steering Group established to provide oversight of all studies supported by the National Institute for Health Research Bristol Cardiovascular Biomedical Research Centre Cardiovascular theme. The text has been updated in section 'Statistical Analysis' on page 14 to provide clarification.

VERSION 2 – REVIEW

REVIEWER	Stephen Froles Sunnybrook Health Sciences Centre University of Toronto Canada
REVIEW RETURNED	07-May-2020

GENERAL COMMENTS	The authors have responded well to almost all the earlier comments. I do not think that the authors answered appropriately to my earlier comment about the 2 level randomization. I suggest that the authors need to accommodate the first level of randomization at least as a sensitivity analysis - I understand why the authors may not want to consider that as a primary analysis. Limitations: I suggest that the authors include a section about limitations. They can mention that the trial may not include some of the highest risks patients, that the intervention may not be appropriate/feasible for some pediatric patients, and that the primary study outcome is a surrogate measure. The authors can describe the rationale for the choice of the primary study outcome. But the method of analysis is looking at serial measurements of high sensitivity between groups, and many of the patients may have findings in the "normal" range of postoperative cardiac biomarkers, the relevance of which is unclear.
--

REVIEWER	Mr Nigel Drury University of Birmingham, UK
REVIEW RETURNED	06-May-2020

GENERAL COMMENTS	Thank you for taking on board my comments and suggestions to improve the manuscript. I have no further requests and now recommend the paper for publication.
--